# Early Prediction of Chronic Kidney Disease: A Comprehensive Performance Analysis of Deep Learning Models

Chaity Mondol [1], F. M. Javed Mehedi Shamrat [2], Md. Robiul Hasan [1], Saidul Alam [1], Pronab Ghosh [3], Zarrin Tasnim [2], Kawsar Ahmed [4,5,*], Francis M. Bui [4] and Sobhy M. Ibrahim [6]

1. Department of Computer Science and Engineering, Daffodil International University, Dhaka 1207, Bangladesh
2. Department of Software Engineering, Daffodil International University, Dhaka 1207, Bangladesh
3. Department of Computer Science, Lakehead University, 955 Oliver Road, Thunder Bay, ON P7B 5E1, Canada
4. Department of Electrical and Computer Engineering, University of Saskatchewan, 57 Campus Drive, Saskatoon, SK S7N 5A9, Canada
5. Group of Bio-Photomatiχ Department of Information and Communication Technology, Mawlana Bhashani Science and Technology University, Tangail 1902, Bangladesh
6. Department of Biochemistry, College of Science, King Saud University, Riyadh 11451, Saudi Arabia
* Correspondence: k.ahmed@usask.ca or k.ahmed.bd@ieee.org or kawsar.ict@mbstu.ac.bd

**Abstract:** Chronic kidney disease (CKD) is one of the most life-threatening disorders. To improve survivability, early discovery and good management are encouraged. In this paper, CKD was diagnosed using multiple optimized neural networks against traditional neural networks on the UCI machine learning dataset, to identify the most efficient model for the task. The study works on the binary classification of CKD from 24 attributes. For classification, optimized CNN (OCNN), ANN (OANN), and LSTM (OLSTM) models were used as well as traditional CNN, ANN, and LSTM models. With various performance matrixes, error measures, loss values, AUC values, and compilation time, the implemented models are compared to identify the most competent model for the classification of CKD. It is observed that, overall, the optimized models have better performance compared to the traditional models. The highest validation accuracy among the tradition models were achieved from CNN with 92.71%, whereas OCNN, OANN, and OLSTM have higher accuracies of 98.75%, 96.25%, and 98.5%, respectively. Additionally, OCNN has the highest AUC score of 0.99 and the lowest compilation time for classification with 0.00447 s, making it the most efficient model for the diagnosis of CKD.

**Keywords:** chronic kidney disease (CKD); OCNN; OANN; OLSTM; Adam; F-measure; precision; sensitivity

---

## 1. Introduction

One of the non-communicable diseases with the quickest growth rate is chronic kidney disease (CKD), a significant cause of death and disease. It has affected more than 10% of the world's population, and millions of people die each year [1]. According to the Global Burden of Disease Study, almost 697.5 million cases of all-stage CKD were registered in 2017, resulting in a global prevalence of 9.1%, up 29.3% from 1990. Meanwhile, between 1990 and 2017, the global all-age death rate from CKD grew by 41.5% [2]. It is predicted that 1 out of every 10 persons has some symptom of kidney disease. It is called a "chronic" disease because kidney disease affects the functioning of the urinary system and develops gradually over time. This illness indirectly impacts global morbidity and mortality rates by increasing the risks of the other main killers (cardiovascular disease, diabetes, hypertension, HIV infection, and malaria). Chronic kidney disease progression is a critical aspect of a person's health. As a result, maintaining excellent kidney function is critical for general health [3].

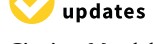



Chronic kidney disease treatment is both expensive and ineffective. In contrast, only about 5% of individuals with early CKD are aware of their condition. Glomerular damage has reached over 50% and is usually irreversible once CKD is identified. In this regard, accurate chronic renal disease prognosis can be highly beneficial. As a result, multiple CKD prediction models for various populations have been developed [4].

In this paper, we proposed three optimized deep learning algorithms, CNN, ANN, and LSTM, to predict CKD at the primary stage. The goal of this classification is to determine whether or not a patient is susceptible to CKD. Multiple layers with linear or nonlinear activation functions are assembled to form the model. These layers are taught to work together to master a complex problem-solving approach [5].

## 2. Related Work

In 2021, Chotimah, et al. [6] introduced a strategy that uses deep learning to identify diagnoses in medical records that contain the most important information about chronic kidney disease. In this study, the first stage is to use the sequential backward feature selection (SBFS) to select a diagnosis that will have a significant impact. Features that do not have as much influence as other features are removed using this algorithm. The top diagnostic features are fed into the artificial neural network (ANN) classification algorithm in the second step. On their study, 18 features are used as input, where 15 features are considered the best features by the system. When 15 diagnostic features were used as inputs, the accuracy of the ANN classification system reached 88%. These results were improved compared to the overall diagnostic features, which have an accuracy of 80%. Alsuhibany, et al. [7] created an IoT-based ensemble of deep-learning-based clinical decision support systems (EDL-CDSS) to diagnose CKD. The adaptive synthetic (ADASYN) technique for the outlier detection process is designed using the EDL-CDSS technique. A convolutional neural network with a gated recurrent unit (CNN-GRU), deep belief network (DBN), and kernel extreme learning machine (KELM) are proposed. They achieved the highest accuracy of 96.91% using the EDL-CDSS approach. Akter, et al. [8], in 2021, deployed seven state-of-the-art deep learning algorithms, ANN, LSTM, GRU, bidirectional LSTM, bidirectional GRU, MLP, and simple RNN, for CKD prediction and classification along with the numerous clinical features of CKD that have been proposed. Their investigation calculated the loss and validation loss in prediction and looked at recall, accuracy, and precision. The ANN, simple RNN, and MLP algorithms performed well in the CKD classification (99%, 96%, and 97% accuracy, respectively). In 2020, Iliyas, et al. [9], using a dataset of 400 patients from Bade General Hospital and 11 features or parameters, used a deep neural network (DNN) to predict chronic kidney disease (CKD). During the preprocessing of the data, several missing cells were simply imputed using the attribute's mean value. They employed the deep neural network (DNN) model to predict if CKD would be present in a patient. The DNN model generated a 98% accuracy rate. Of the 11 variables, creatinine and bicarbonate impact CKD prediction most. In 2020, Ma, et al. [10] suggested chronic kidney illness utilizing a heterogeneous modified artificial neural network based on deep learning. On the Internet of Medical Things (IoMT) platform, they used a heterogeneous modified artificial neural network (HMANN) for the early detection, segmentation, and diagnosis of chronic renal failure. To identify the location of kidney stones, HMANN assists in reducing noise and segmenting the kidney image. The accuracy rates for ANN-SVM and HMANN were 92.3% and 97.5%, respectively. A bidirectional long short-term memory (LSTM) network and a one-dimensional correlational neural network (1-D CorrNN) are combined in the deep learning model presented by Bhaskar, et al. [11], in 2020. The suggested model is trained and evaluated using the CKD-sensing module. The accuracy of the suggested model, 1-D CorrNN-LSTM, is 98.08%, and it performed better than other models. In 2019, N. Bhaskar, et al. [12] proposed an effective and novel method for non-invasive chronic renal disease diagnosis. The proposed task involves designing and building a novel sensing module for CKD diagnostics. They used a hybrid deep learning convolution neural network–support vector machine (CNN-SVM)

model to make predictions. The proposed model is put to the test in experiments, and its performance is compared to that of a traditional CNN. The accuracy of the suggested hybrid deep learning model (CNN-SVM) was 96.59%. In 2019, Almansour, et al. [13] used classification techniques including a artificial neural network (ANN) and a support vector machine (SVM). Using the mean of the corresponding attributes, they replaced all missing values in the datasets. Additionally, they employed a 10-fold cross-validation procedure to divide the training and test datasets according to the ratio (90:10). In their proposed method, ANN performs better. Using the optimized features, the accuracy is 99.75%, while, from SVM, the accuracy is 97.75%.

## 3. Materials and Methods

This study intends to test various classification algorithms and select the most crucial ones that can provide patients with chronic kidney disease with the greatest diagnostic support. To do this, an optimized proposed model for recognizing diseases that is essential for categorizing patients' characteristics, disease prevalence, and hygienic circumstances is offered. The following methodological approaches are used on the clinical data described in the next sections.

1.  Performing data preprocessing to confirm accuracy through the detection of extreme situations, removing noisy data and missing values.
2.  Choosing the best classifier, by contrasting regularly used classification methods with CKD studies from the literature review and ablation study.
3.  An optimized model based on CNN architecture is proposed.
4.  The precision, recall, specificity, and *F*1 score are calculated to support the model accuracy. The effectiveness of the models is evaluated using the loss function as well.
5.  The AUC value is computed in order to assess the proposed model.

Our suggested approach is built on kidney disease datasets. We split our dataset into train and test (80% data on train and 20% data on test) and showed that the model was free of overfitting issues. All the classifiers introduced were designed and obtained the best accuracy from the dataset. Figure 1 presents the complete aspects of our approach.

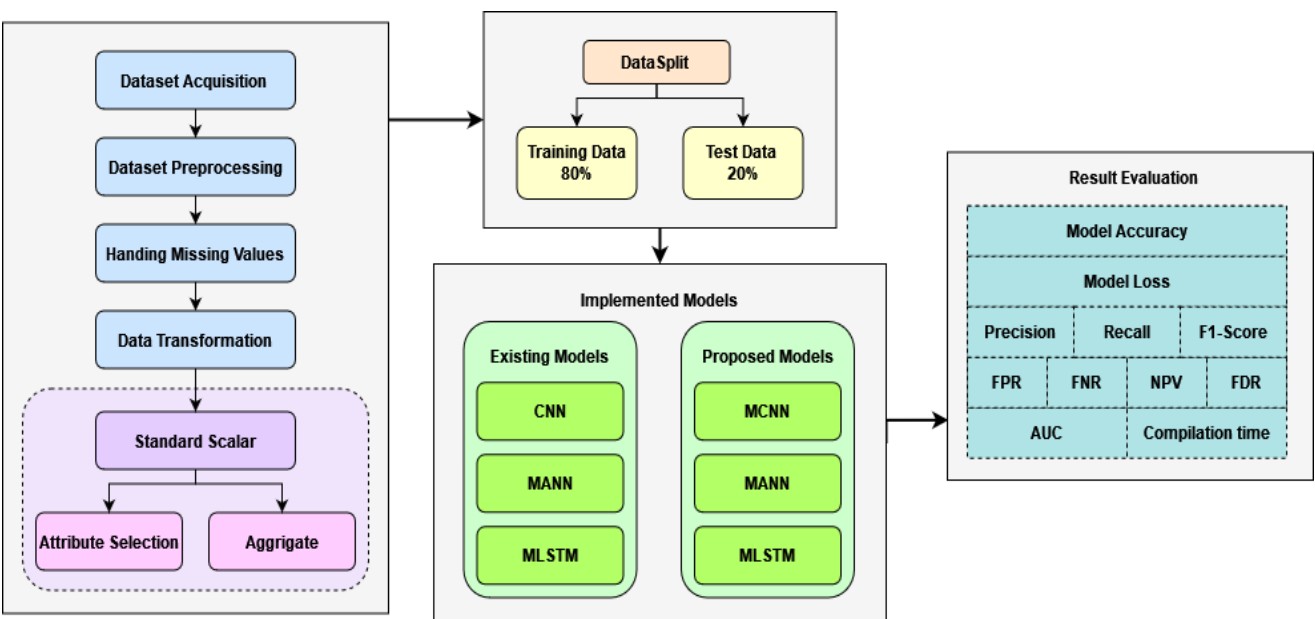

**Figure 1.** The proposed system architecture for CKD prediction.

### 3.1. Study Data

The UCI machine learning repository's chronic kidney disease dataset was used in this study. The dataset has 400 records, each composed of a set of 25 attributes [14]. The

'classification' variable indicates whether the patient has CKD or not. This variable is preserved as a dependent or target variable during the classification process. The rest variables are fed as input to the classifier model to predict the target class. The type for the variables Age, BP, Bgr, Bu, Sc, Sod, Pot, Hemo, Pcv, Wc, and Rc is numerical, whereas the variables Sg, Al, Su, Rbc, Pc, Pcc, Ba, Htn, Dm, Cad, Appet, Pe, Ane, and classification are nominal in type. Our target variable (classification) has two categories of nominal value (ckd and not ckd). To develop our proposed approach, we mapped this value into numerical values (0, 1).

There are 400 samples in the collection, each with 25 attributes. There are 250 samples in the CKD category, and 150 samples in the Not CKD category, among the available samples. Table 1 depicts the characteristics.

**Table 1.** Data variables for patients' CKD dataset description.

| S. No | Attribute | Datatype | Description | Unit of Measurement |
|-------|-----------|----------|-------------|---------------------|
| 1 | Age | Numerical | Age | Years |
| 2 | bp | Numerical | Blood Pressure | mm/Hg |
| 3 | sg | Nominal | Specific Gravity | 1.005, 1.010, 1.015, 1.020, 1.025 |
| 4 | al | Nominal | Albumin | 0, 1, 2, 3, 4, 5 |
| 5 | su | Nominal | Sugar | 0, 1, 2, 3, 4, 5 |
| 6 | rbc | Nominal | Red Blood Cells | Normal, Abnormal |
| 7 | pc | Nominal | Pus Cell | Normal, Abnormal |
| 8 | pcc | Nominal | Pus Cell Clumps | Present, Not Present |
| 9 | ba | Nominal | Bacteria | Present, Not Present |
| 10 | bgr | Numerical | Blood Glucose Random | mgs/dL |
| 11 | bu | Numerical | Blood Urea | mgs/dL |
| 12 | sc | Numerical | Serum Creatinine | mgs/dL |
| 13 | sod | Numerical | Sodium | mEq/L |
| 14 | pot | Numerical | Potassium | mEq/L |
| 15 | hemo | Numerical | Haemoglobin | gms |
| 16 | pcv | Numerical | Packed Cell Volume | 0, 1, 2, . . . |
| 17 | wbcc | Numerical | White Blood Cell Count | cells/cumm |
| 18 | rbcc | Numerical | Red Blood Cell Count | millions/cumm |
| 19 | htn | Nominal | Hypertension | Yes, No |
| 20 | dm | Nominal | Diabetes Mellitus | Yes, No |
| 21 | cad | Nominal | Coronary Artery Disease | Yes, No |
| 22 | appet | Nominal | Appetite | Good, Poor |
| 23 | pe | Nominal | Pedal Edema | Yes, No |
| 24 | ane | Nominal | Anemia | Yes, No |
| 25 | Class | Nominal | CKD, Not CKD | CKD, Not CKD |

*3.2. Dataset Preprocessing*

After completing the data collection, it is vital to focus on any missing values in the dataset. The prediction efficiency will be affected if missing values are present. In our dataset, there are some missing values. We applied numerous imputations (MI) to fill up the missing variables. A missing values is handled in our approach by replacing it with the mean or average value of the considered attribute. As a result, more accurate and actual forecast results will be obtained. The current nominal variables are then transformed to numerical values ranging from 0 to 1. These processes will aid in the acquisition of a preprocessed dataset. Any classifier model can now be fit to this dataset.

In our view, Figure 2 represents correlated features with the predicted class attribute (classification). The attribute values define the strength of the correlated features at the right portion (range from $-0.6$ to 0.6), in accordance with the lightness of color. The Figure represents 'pcv' and 'rc' as having a strong correlation with 'htn', having the value of 0.74, 0.68; whereas 'sod','htn' has a lesser correlation with 'hemo'. having the value of $-0.62$, $-0.5$ approximately.

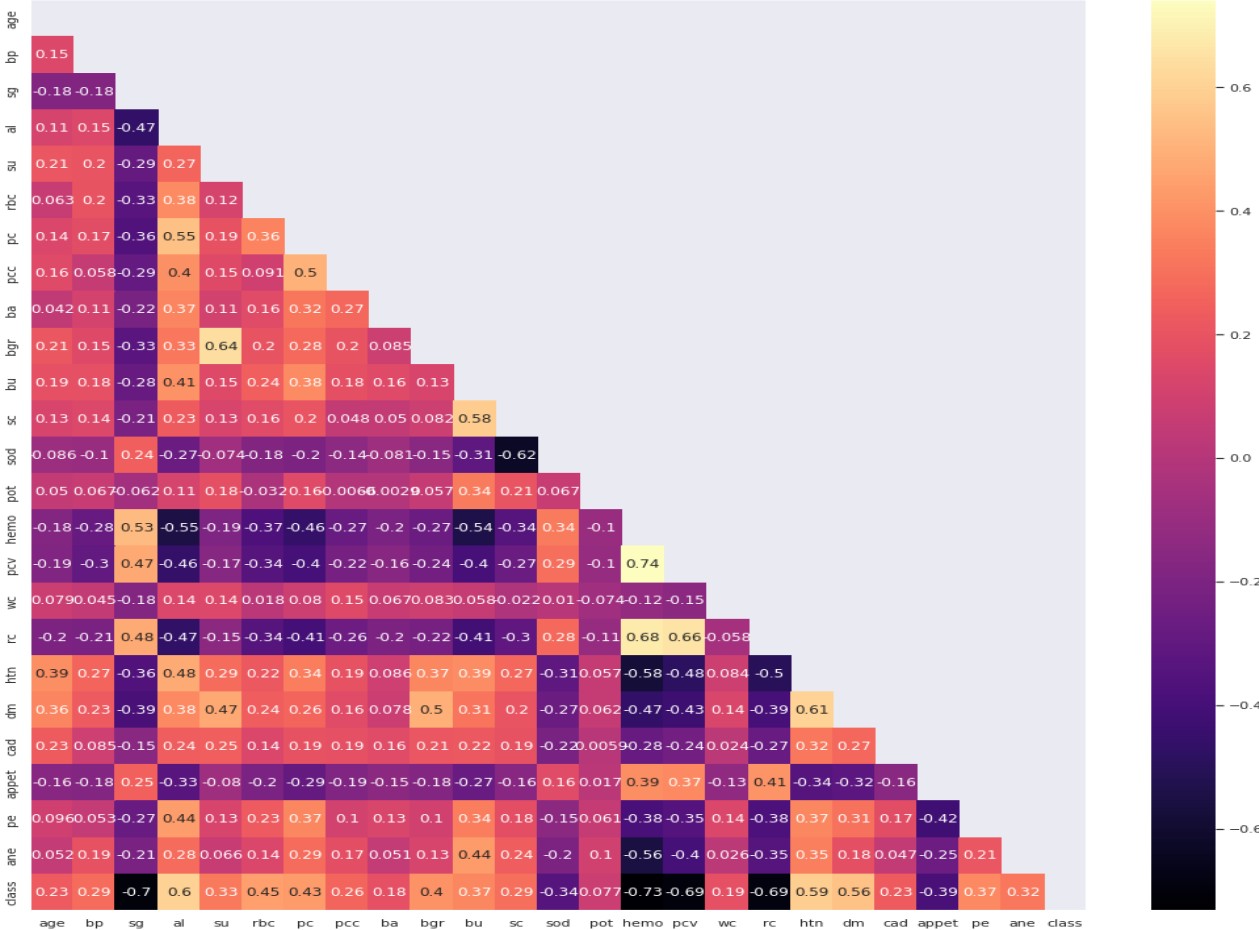

**Figure 2.** Highly correlated features of CKD dataset.

*3.3. Proposed Methodology*

The implementation of optimized CNN, ANN, and LSTM classifiers is explained in this section. To obtain the training and testing datasets, the preprocessed data is divided in an 80:20 ratio. The training dataset is used to fit the classifier models, and the testing dataset is used to collect predictions. The optimized CNN classifier is implemented with a kernel regularization parameter of C = 1.0 and the activation function is ReLu. For the greatest known performance, the LSTM classifier is learned at a rate of 1.0 and with 400 estimators. The predicted performance of these classifiers will be used to justify the proposed classifier's performance.

Table 2 lists the parameters that were used in our model. Here, we can see the parameters (P), activation layer (AL), optimizer (OP), learning rates (LR), epochs (E), and compilation time (CT). In our study, we use sigmoid and ReLu as an activation layer, and Adam and Adamax as optimizer, to acquire good accuracy. The using activation layer, optimizer, and learning rates fit our study, and our model produced good accuracy.

**Table 2.** Different algorithms' parameters.

| Model | P | E | AL | OP | LR | CT |
|---|---|---|---|---|---|---|
| Optimized CNN | 6497 | 8 | ReLu, sigmoid | Adam | 0.1 | 0.00968 |
| Optimized ANN | 381 | 8 | ReLu, sigmoid | Adam, Adamax | 0.01 | 0.00447 |
| Optimized LSTM | - | 8 | ReLu | Adam | - | 0.00527 |

### 3.4. Optimized Convolutional Neural Network (OCNN)

We proposed an optimized convolutional neural network (OCNN) for the early detection of chronic kidney disease. Since the early 21st century, the CNN algorithm has been used to classify signals [15]. Although the OCNN method beats traditional machine learning approaches in many situations, deep learning techniques can improve its performance even further. Figure 3 shows the architecture diagram for our OCNN model, where two convolutional layers, with two dense layers and one output layer, are available. The first and second convolutional layers have dense = 16, kernel_regularizer = 0.01, dropout = 0.1, and activation type = ReLu. In the third layer, dense 1 also has dense = 16, kernel_ regularizer = 0.01, dropout = 0.1, and activation type = ReLu. In the fourth layer, dense 2 has dense = 8, kernel regularizer = 0.01, dropout = 0.15, and activation type = ReLu. Finally, the output layer has dense = 1 and an activation layer with a sigmoid.

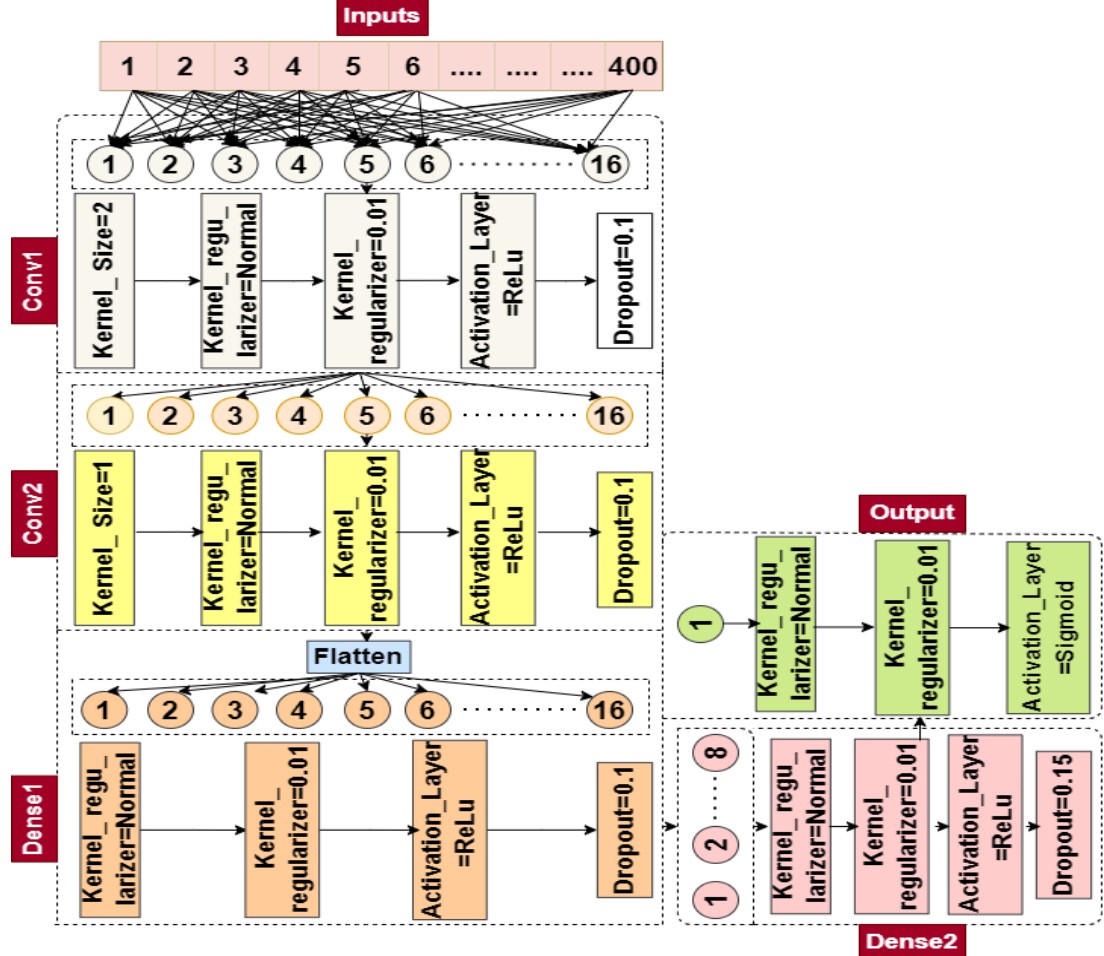

**Figure 3.** The architecture of optimized convolutional neural network (OCNN), a fine-tuned network.

### 3.5. Ablation Study of the Proposed Model (OCNN)

As part of an ablation study, three experiments were carried out by changing various elements of the proposed OCNN model. It is possible to create a more robust design with improved classification accuracy by changing various components. The following features were conducted in an ablation study: changing the convolution and dense layer, changing the activation layer, and dropout.

#### 3.5.1. Ablation Study 1: Changing Convolution Layer and Dense Layer

In this study, Table S1 show the train accuracy and validation accuracy of the different types of convolution layers and dense layers. Convolution layer 2 and dense layer 3

achieved the highest accuracy, with a train accuracy of 96.88% and a validation accuracy of 98.75%. However, changing the convolution layer and dense layer, we obtained a modest accuracy for the train accuracy, 98.44%, 97.81%, and 95.63%, respectively, and the validation accuracy is 86.25%, 95%, and 95%, respectively. The accuracy decreases in convolution layers 6 and 7. Moreover, in dense layers 4 and 5, they are 94.06%, 67.50% and 61.87%, 65%, respectively.

### 3.5.2. Ablation Study 3: Changing the Activation Function

In Table S2, activation function PReLu, leaky ReLu, sigmod, tanh, and ReLu are applied to find the best results for the train accuracy and validation accuracy. We obtained the highest accuracy from activation function ReLu. Using ReLu, we obtained a train accuracy of 96.88% and a validation accuracy of 98.75%. Applying activation function PReLu, leaky ReLu, sigmod, and tanh, the results dropped and achieved 61.87%, 63.75%, 95.63%, and 89.38% for the train accuracy, respectively. The validation accuracy for PReLu, leaky ReLu, sigmod, and tanh is, respectively, 65%, 68.75%, 77.50%, and 92.50%.

### 3.5.3. Ablation Study 4: Changing the Dropout Value

In this study, we changed the dropout value to improve the training accuracy and validation accuracy, as depicted in Table S3. The highest accuracy was achieved from a dropout value of 0.1, with a train accuracy of 96.88% and a validation accuracy 98.75%. However, when changing the dropout value, the accuracy decreases. For a dropout value of 0.2, 0.15, and 0.05, the train accuracy is 95.63%, 97.50%, and 98.12%, respectively, and the validation accuracy is 88.75%, 90%, and 92.50%, respectively.

### 3.6. Optimized Artificial Neural Network (OANN)

Artificial neural network models have been identified as a promising candidate for pattern classification problems [16]. Parallel computing systems called artificial neural networks were developed with the intention of developing a digital model of the human brain. To design a system that is both faster and more accurate than current traditional techniques is the main driver behind the development of an ANN. In an artificial neural network, each node is linked to every other node, allowing information to pass from one node to the next until it reaches the output layer, where the error is calculated and reduced to increase the model's productivity [17]. We proposed an optimized ANN (OANN) model in our study. Figure 4 shows the architecture diagram for our optimized ANN network, where three input layers and one output layer are available. The first input layer has 8 units, and the kernel_regularizer = 0.03. For both the second and third layers, the value of the kernel_regularizer is 0.01, and the optimizer is Adam and Adamex and the activation layer is ReLu and sigmoid.

### 3.7. Ablation Study of the Proposed Model (Optimized ANN)

Four tests were run by altering different components of the suggested optimized ANN model, as part of an ablation investigation. It is possible to produce a more reliable design with increased classification accuracy, by altering different components. The following elements conducted an ablation study: changing dense layer, changing activation function, and changing kernel_initializer and optimizer.

### 3.7.1. Ablation Study 1: Changing Dense Layer

In this work, Table S4 displays the train and validation accuracy of several dense layer types. With a training accuracy of 98.12% and a validation accuracy of 96.25%, dense layer 3 had the best accuracy. However, when the dense layer was changed, the validation accuracy of the suggested model dropped to 61.87%, 71.48%, and 65.00%.

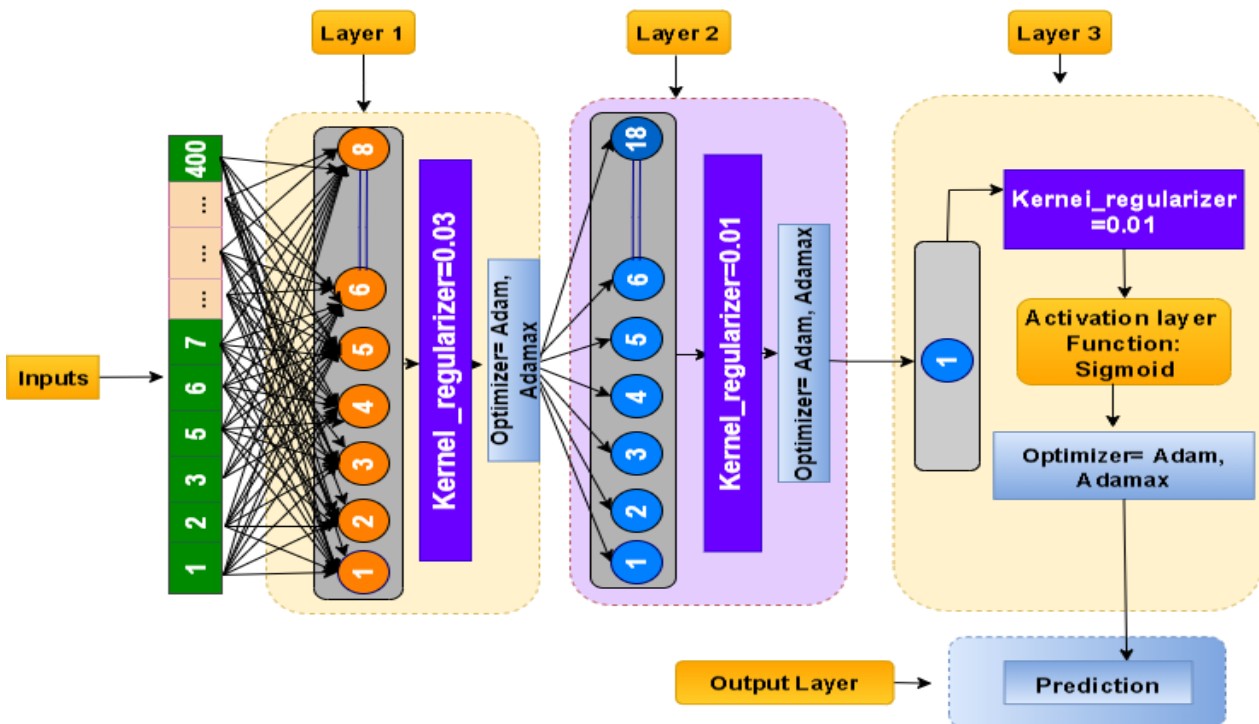

**Figure 4.** The architecture of optimized artificial neural network (OANN), a fine-tuned network.

### 3.7.2. Ablation Study 2: Changing the Activation Function

Table S5 compares the best train accuracy and validation accuracy results using the activation functions PReLu, leaky ReLu, sigmod, tanh, and ReLu. The activation function ReLu gave us the maximum accuracy, with a training accuracy of 98.12% and a validation accuracy of 96.25%. We also applied theg activation function leaky ReLu, PReLu, sigmod, and tanh and obtained results for the train accuracy of 38.12%, 40.62%, 61.87%, and 70.94%, respectively, as well as for the validation accuracy of 35%, 38%, 65%, and 71.25%, respectively.

### 3.7.3. Ablation Study 3: Changing Kernel Initializer

A number of kernel initializers were investigated to obtain the best train and validation accuracies for the new proposal. The numerous types of kernel initializers are shown in Table S6. Applying the standard as the kernel initializer gave us the best accuracy, with a train accuracy of 98.12% and a validation accuracy of 96.25%. When we utilized He normal and normal as kernel initializers, the accuracy decreases by 90.62% and 93.44% for train accuracy, respectively, and 92.50% and 95% for validation accuracy, respectively.

### 3.7.4. Ablation Study 4: Changing the Optimizer

In Table S7, Nadam, RMSprop, Adamax, and Adam are applied as the optimizer in our proposed model, to find the best results for the train accuracy and validation accuracy. We obtained the highest accuracy from the Adam optimizer. Using Adam, we obtained a train accuracy of 98.12% and a validation accuracy of 96.25%. The accuracy dramatically dropped for the application of the optimizers Nadam, RMSprop, and Adamax. The train accuracy is 61.87%, 92.50%, and 97.19%, respectively, and, similarly, the validation accuracy is 65%, 95%, and 95%, respectively.

### 3.8. Optimized Long Short-Term Memory (OLSTM)

In our work, we proposed an optimized long short-term memory (LSTM). It is one of the most prominent RNN variations that uses a gated architecture to handle sequential data. In health informatics, LSTM has lately been used with encouraging results. Standard

LSTM networks, on the other hand, have a limited ability to deal with temporal irregularity and dependencies [18]. In Figure 5, we can see the architecture diagram for an OLSTM network, where four dense layers and one output layer are available. The dense layers are, correspondingly, 64, 32, 24, and 64. In the first and second dense layer the kernel_regularizer is 0.03 and 0.001, respectively, and the activation layer used ReLu for both of them. The dropout for all dense layers, respectively, are 0.5, 0.1, 0.08, 0.8.

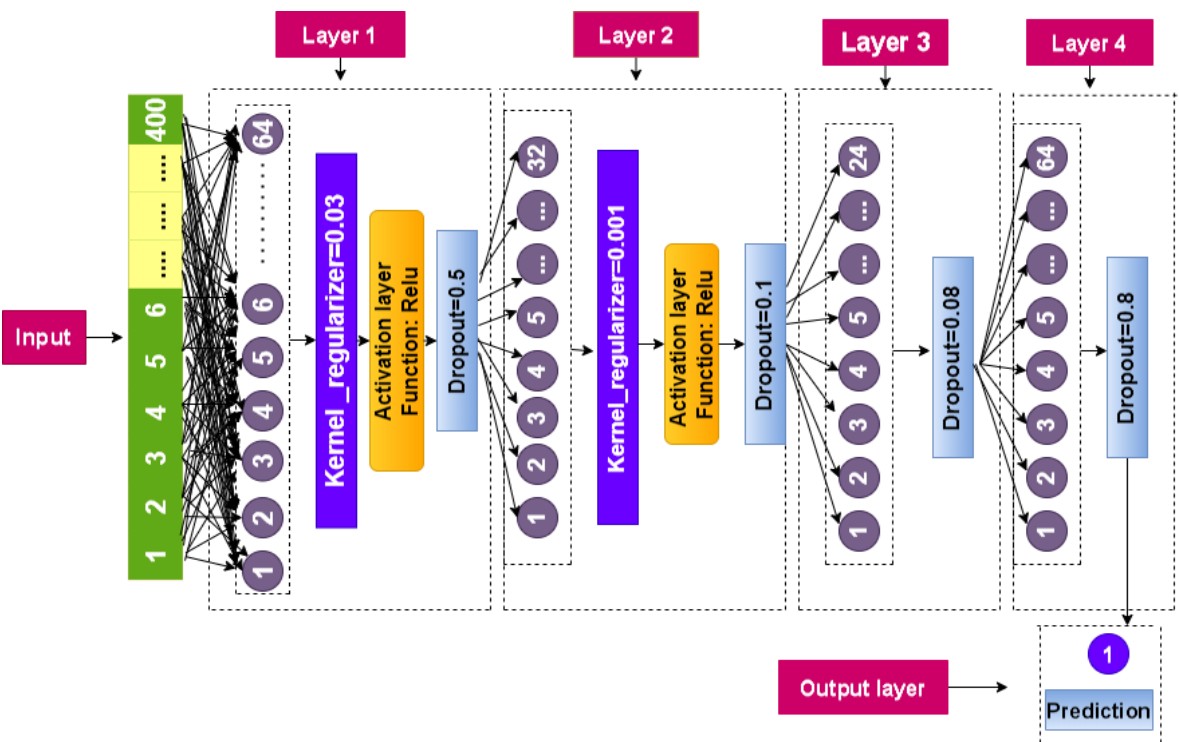

**Figure 5.** The architecture of optimized long short-term memory (OLSTM), a fine-tuned network.

*3.9. Ablation Study of the Proposed Model (OLSTM)*

We performed four experiments as part of an ablation investigation, by altering different parts of the suggested improved LSTM model. It is feasible to produce a more reliable design with increased classification accuracy by altering different components. Changes were made to the LSTM layers, dense layers, dropout layers, activation function, kernel initializer, and optimizer during the ablation investigation.

### 3.9.1. Ablation Study 1: Changing LSTM, Dense, and Dropout Layers

Table S8 in this study displays the train accuracy and validation accuracy of various LSTM, dense, and dropout layer types. The four LSTM, one dense, and four dropout layers yielded the best accuracy, with a training accuracy of 97.81% and a validation accuracy of 97.50%. However, when changing the layers, the accuracy of the proposed model dropped for the train accuracy to, respectively, 57.16%, 65%, 64.38%, and 75%, while the validation accuracy is, respectively, 61.25%, 77.50%, 80%, and 82.50%.

### 3.9.2. Ablation Study 2: Changing the Activation Function

In Table S9, activation functions PReLu, leaky ReLu, sigmod, tanh, and ReLu are applied to find the best results for the train accuracy and validation accuracy. We obtained the highest train accuracy of 97.81% and validation accuracy of 97.50% from the activation function ReLu. For the activation function sigmod, leaky ReLu, PReLu, and tanh, the results significantly dropped, with 51.88%, 61.56%, 75%, and 77.19% for train accuracy, respectively. The validation accuracy for sigmod, leaky ReLu, PReLu, tanh is 65%, 70%, 82.50%, and 82.50%, respectively.

### 3.9.3. Ablation Study 3: Changing Kernel Initializer

To identify the best accuracy for the new proposal, a variety of kernel initializers were explored. Table S10 shows the different type of kernel initializer for finding the ideal train accuracy and validation accuracy. We obtained the highest accuracy by applying the uniform as the kernel initializer, with a train accuracy of 97.81% and a validation accuracy of 97.50%. The accuracy decreases when we use He normal and normal as the kernel initializer, obtaining the accuracy of 82.81% and 71.56%, respectively, for the train accuracy, and 37.50% and 78.75%, respectively, for the validation accuracy.

### 3.9.4. Ablation Study 4: Changing the Optimizer

In Table S11, Nadam, RMSprop, Adamax, and Adam are applied as optimizers in our proposed model, to find the best results for the train accuracy and the validation accuracy. We obtained the highest accuracy from the Adam optimizer. Using Adam, we obtained a train accuracy of 97.81% and a validation accuracy of 97.50%. Besides, optimizer Nadam, RMSprop, and Adamax achieved the training accuracy of 66.87%, 75.63%, and 76.56%, respectively, and the validation accuracy of 66.25%, 77.50%, and 82.50%, respectively.

### 3.10. Evaluation Process

The evaluation technique utilizes the accuracy score, precision, recall, and *F*1 score [19]. Accuracy defines the rate of correct predictions by the model. While precision denotes the quality of positive prediction, recall determines the model's ability to detect relevant data. *F*1 score determines the harmonic mean value between recall and precision.

$$Accuracy = \frac{TP + TN}{TP + TN + FP + FN} \tag{1}$$

$$Precision = \frac{TP}{TP + FP} \tag{2}$$

$$Recall = \frac{TP}{TP + FN} \tag{3}$$

$$F1 - Score = 2\left(\frac{Precision \times Recall}{Precision + Recall}\right) \tag{4}$$

The false positive rate (*FPR*) computes the model's incorrect predictions of positive values contrary to the false negative rate (*FNR*), determining the incorrect predictions of negative values. The negative predict value (*NPV*) determines which negative prediction is in fact negative. The false discovery rate (*FDR*) measures the number of false predictions determined as true.

$$FPR = \frac{FP}{FP + TN} \tag{5}$$

$$FNR = \frac{FN}{FN + TP} \tag{6}$$

$$NPV = \frac{TN}{TN + FN} \tag{7}$$

$$FDR = \frac{FP}{FP + TP} \tag{8}$$

Here, true positive (*TP*) means that values classified as true in theory are also true in reality. False positive (*FP*) occurs when false results are incorrectly labeled as true. False negative (*FN*) denotes a value that is positive but incorrectly recognized as negative. When a value is correct but incorrectly labeled as negative, it is designated as true negative (*TN*).

## 4. Experiment and Results

Our proposed three optimized deep learning models, OCNN, OANN, and OLSTM-based CKD prediction, are reported in this work. We examined each model's accuracy, sensitivity, recall, and *F*1 score.

In Table 3, the classification result of the proposed optimized convolution neural network (OCNN), optimized artificial neural network (OANN), and optimized long short-term memory (OLSTM) is explained. From optimized CNN, we obtained the highest accuracy of 98.75%. From the optimized ANN and optimized LSTM, we obtained 96.25% and 97.50% accuracy, respectively. The precision of OCNN is 96.55%, OANN is 90.32%, and OLSTM is 93.33%. When compared to existing classifiers, the results were also superior in terms of specificity, precision, and accuracy. The *F*1 score of OCNN is 99%, OANN is 97%, and OLSTM is 98%. The deep learning models (OCNN, OANN, and OLSTM) gave the value of recall of 98%, 94%, and 96%, respectively.

**Table 3.** Performance comparison of proposed models.

| Proposed Model | Performance Measure | | | |
| --- | --- | --- | --- | --- |
| | **Accuracy** | **Recall** | **Precision** | ***F*1-Score** |
| Optimized CNN | 98.75% | 98% | 96.55% | 99% |
| Optimized ANN | 96.25% | 94% | 90.32% | 97% |
| Optimized LSTM | 97% | 96% | 93.33% | 98% |

### 4.1. Training Accuracy of the Models

Figure S1 shows the accuracy of implemented models for eight epochs. Furthermore, a comparison of the optimized models are shown with the existing CNN, ANN, and LSTM models. The training's accuracy of the OCNN starts at 98.12% and ends at 96.88%. For OANN training, accuracy ranges from 97.19% to 98.12%, and the training accuracy of OLSTM starts at 92.50% and ends at 97.81%.

Compared to the optimized models, the existing CNN, ANN, and LSTM models have much lower performance accuracy overall. For CNN, ANN, and LSTM, the training accuracy ranges from 88.37% to 90.4%, 90.88% to 88.72%, and 82.15% to 91.22%, respectively.

### 4.2. Validation Accuracy of the Models

Figure S2 displays the validation accuracy of the OLSTM that starts from 96.25% to 98.5% after eight epochs. The accuracy of validation drops drastically in epoch three but raises again in epoch four. For OCNN and OANN, the accuracy increases from 97.5% to 98.75% and from 98.75% to 96.25%, respectively. The highest accuracy of CNN, ANN, and LSTM remained comparatively low, at 92.71%, 90.43%, and 88.51%, respectively.

### 4.3. Training Loss of the Models

Figure S3 displays the training loss of the models. In this illustration, the training loss of OCNN model starts at 0.2668 and ends at 0.2401 after eight epochs. For OANN, the training model loss starts at 0.2238 and ends at 0.2354. The training loss of OLSTM starts at 0.8814 and ends at 0.1945. The lowest training loss of the models CNN, ANN, and LSTM is 0.3756, 0.3334, and 0.3261, respectively, for epoch eight.

### 4.4. Validation Loss of the Models

Figure S4 shows the validation loss of models after eight epochs. The validation loss of OANN ranges from 0.1808 to 0.2425. For OCNN, the validation loss ranges from 0.2943 to 0.1998. The validation loss of OLSTM ranges from 0.6917 to 0.1876. The models CNN and LSTM have the lowest validation loss of 0.3558 and 0.4052, respectively, for epoch eight. However, ANN has the lowest loss of 0.4207, in epoch seven.

### 4.5. Area under Curve (AUC) Values of the Used Models

In Table 4, we depicted our proposed model, where the OCNN model gives the highest AUC score, of 0.99, and both OANN and OLSTM achieved 0.97. Besides, CNN, ANN, and LSTM obtained the AUC score of 0.88, 0.83, and 0.81, respectively.

**Table 4.** Experimental result of AUC score among the models.

| | Models | | | | | |
|---|---|---|---|---|---|---|
| **Performance Measure** | **OCNN** | **OANN** | **OLSTM** | **CNN** | **ANN** | **LSTM** |
| AUC Score | 0.99 | 0.97 | 0.97 | 0.88 | 0.83 | 0.81 |

### 4.6. Runtime Analysis

Figure 6 shows the runtime of the classifiers in seconds. It is seen that the OCNN model performance has the lowest time, of 0.00447 s. The highest compilation time is received from the OANN classifier, of 0.00968 s. The classifier OLSTM performs with the time of 0.00527 s.

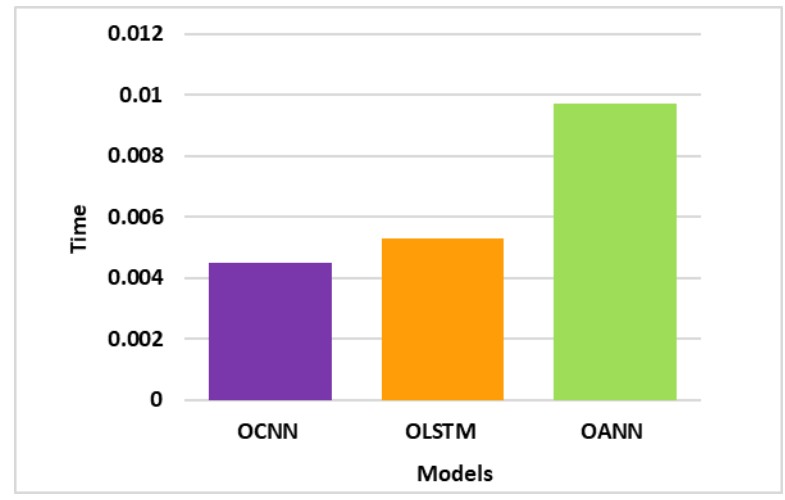

**Figure 6.** Runtime analysis among the models.4.7. Evaluation of Prediction Result.

Table 5 contains the false positive rate, false negative rate, negative predictive value, and false discovery rate for the three classifiers.

**Table 5.** Evaluation of prediction result.

| **Prediction Result** | **OCNN** | **OANN** | **OLSTM** |
|---|---|---|---|
| False positive rate | 0.0 | 0.0 | 0.0 |
| False negative rate | 3.448 | 9.677 | 6.666 |
| Negative predictive value | 98.07% | 94.23% | 96.16% |
| False discovery rate | 0.0 | 0.0 | 0.0 |

### 4.7. Discussion

In the literature, some hybrid model, cross validation, and feature selection techniques were used, which are missing in our study. A greater memory and longer compilation time are required for such complicated techniques [20]. To reduce compilation time of a model, we avoid a cross-validation technique or a hybrid model. Dropout, l2 regularization, and dense layers can be used to handle overfitting issues [21]. Instead of cross-validation techniques, we proposed our model with some balanced dense layers, and kernel regularizers are added to some neurons. Then, we disable a few parameters in order to balance our data. Issues with overfitting are handled via kernel regularizers and dropouts. Finally, we

achieve a balanced performance from all the different optimizers, and our model generates the findings in a very short time. In Table 6, the proposed model is compared with other models, including ANN, ensemble of deep learning based clinical decision support systems (EDL-CDSS), simple RNN, DNN, ANN-SVM, 1-D corrNN-LSTM, and SVM. The table displays the experimental results when the proposed model was tested. Our proposed model provides better accuracy, compared with the existing models.

**Table 6.** Comparison of proposed model with existing work.

|  | Author | Model | Accuracy |
| --- | --- | --- | --- |
| Existing work | Chotimah, et al. [6] | ANN | 88% |
|  | Alsuhibany, et al. [7] | EDL-CDSS | 96.91% |
|  | Akter, et al. [8] | Simple RNN | 96% |
|  | Iliyas, et al. [9] | DNN | 98% |
|  | Ma, et.al. [10] | ANN-SVM | 92.3% |
|  | Bhaskar, et al. [11] | 1-D CorrNN-LSTM | 98.08% |
|  | N. Bhaskar, et al. [12] | CNN-SVM | 96.59% |
|  | Almansour, et.al. [13] | SVM | 97.76% |
| Our Work | - | Optimized ANN | 96.25% |
|  | - | Optimized LSTM | 97% |
|  | - | Optimized CNN | 98.75% |

## 5. Conclusions

A neural-network-based method for detecting chronic kidney disease (CKD) has been successfully developed here. This prediction technique has a high level of accuracy and may be used by physicians as an alternative approach. It can also be used by ordinary people to determine the likelihood of developing CKD. Using a neural network, this study effort provides an efficient and cost-effective solution to the CKD-detection issue. In the study, by filling in the most likely values for all missing variables, the raw dataset is transformed into a highly preprocessed dataset. All the learning algorithms, optimized and traditional, were evaluated on the same dataset and on the heavily preprocessed data, where the optimized models, OCNN, OANN, and OLSTM, performed well. Based on prediction accuracy, OCNN was determined to be the best algorithm for the kidney dataset with an AUC score of 0.99 and a compilation time of 0.00447 s. From our overall work, we obtained the highest accuracy rate from OCNN, with 98.75%, followed by OLSTM and OANN, with 97% and 96.25%, respectively. Contrary to the high accuracy of the optimized models, the traditional models had unsatisfactory performance, with 92.71%, 90.43%, and 88.51%, for CNN, ANN, and LSTM, respectively. Our future research will concentrate on creating detection methods for additional illnesses and evaluating the effectiveness of neural networks on those diseases using various learning algorithms. Our primary goal is to enhance the detection system for illnesses, particularly chronic and severe disorders. In the future, we will work with merged datasets, where we will implement some deep learning and machine learning hybrid model. Due to having such complex architecture, the models tend to do overfitting. To handle this issue, we will use k-fold cross validation and feature-selection techniques. In our future work, variance results will be measured from the models, and we will check the validity of our proposed models on different datasets. In addition, we will also apply statistical significance testing on our proposed models.

**Supplementary Materials:** The following supporting information can be downloaded at: https://www.mdpi.com/article/10.3390/a15090308/s1, Figure S1: Comparison of training accuracy of the models; Figure S2: Comparison of validation accuracy of the models; Figure S3: Comparison of training loss of the models; Figure S4: Comparison of validation loss of the models; Table S1: Changing convolution layer and dense layer are to evaluate the ablation study; Table S2: Changing the activation function to evaluate the ablation study; Table S3: Changing the dropout value to evaluate the ablation study; Table S4: Changing dense layers to evaluate the ablation study; Table S5: Changing the activation function to evaluate the ablation study; Table S6: Changing kernel initializer

to evaluate the ablation study; Table S7: Changing the optimizer to evaluate the ablation study; Table S8: Changing dense layers to evaluate the ablation study; Table S9: Changing the activation function to evaluate the ablation study; Table S10: Changing kernel initializer to evaluate the ablation study; Table S11: Changing the optimizer to evaluate the ablation study.

**Author Contributions:** Conceptualization, F.M.J.M.S. and K.A.; methodology, C.M., F.M.J.M.S., M.R.H., S.A., P.G. and Z.T.; software, F.M.J.M.S., K.A. and Z.T.; validation, F.M.J.M.S., P.G., F.M.B., S.M.I. and K.A.; formal analysis, C.M., F.M.J.M.S., M.R.H., S.A., P.G. and Z.T.; investigation, C.M., F.M.J.M.S., M.R.H., S.A., P.G., Z.T. and K.A.; resources, F.M.J.M.S., P.G. and K.A.; data curation, F.M.J.M.S. and P.G.; writing—original draft preparation, C.M., F.M.J.M.S., M.R.H., S.A., P.G. and Z.T.; writing—review and editing, F.M.B., S.M.I. and K.A.; supervision, F.M.J.M.S., F.M.B. and K.A.; project administration, K.A.; funding acquisition, F.M.B. and S.M.I. All authors have read and agreed to the published version of the manuscript.

**Funding:** This work was supported in part by funding from the Natural Sciences and Engineering Research Council of Canada (NSERC).

**Institutional Review Board Statement:** Not applicable.

**Informed Consent Statement:** Not applicable.

**Data Availability Statement:** The datasets used for this study can be obtained from the source mentioned in Reference [14]. The complete set of code can also be obtained from: https://github.com/robiulxasan/Kidney-Disease.git (accessed on 5 April 2022).

**Acknowledgments:** This work was supported by Researchers Supporting Project number (RSP-2021/100), King Saud University, Riyadh, Saudi Arabia.

**Conflicts of Interest:** The authors declare no conflict of interest.

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
