# Peer review of "Early Prediction of Chronic Kidney Disease: A Comprehensive Performance Analysis of Deep Learning Models"

_algorithms, doi:10.3390/a15090308_

Round 1

Reviewer 1 Report

The presented manuscript shows some results related to deep learning models for Chronic Kidney Disease prediction. The following concerns need to be addressed:

Major comments:
1) The datasets, ML models, and raw predictions used/presented in this study had not been provided. Without them, it is impossible to validate any of the presented results. Please provide them as supplementary data or/and as a Github repository.

2) Extensive editing of English language and style required. The text is full of typos (l.249, 250, 253, etc.), and grammar errors (e.g. "Alsuhibany et al., [7] 2021 In this study, they"). The whole text needs to be rewritten, it contains multiple fragments that are badly organized, and some sentences are not needed (trivial information) or are very similar to the previous ones. Some sentences make no sense. For instance, lines 130-136 are hard to read, and the sentence "The dataset is processed to 131 provides a better intuition." make no sense. Then we have repetition related to 80% and 20 % vs 80:20.

3) Use proper nomenclature e.g in computer science we do not use compilation time but rather runtime in the context presented in the manuscript

4) I do not understand the ML section as it is described now. I do not know in what way the so-called modified versions of CNN, ANN and LSTM were different than the standard ones. The text describes very standard optimization of the number of layers or/and hyperparameters. I do not see here a novel modification of the classical CNN, ANN, or LSTM architectures. The whole description of ML models is very poor. For instance, I do not understand what the RBF kernel has to do to CNN (l.177-180). In figure 3 regarding the CNN network, there is only some MLP model presented (lack of CNN elements). In general, you need to compare your best, optimized versions of CNN, ANN, and LSTM models to the ones designed by others (some of the methods mentioned in the "Related work" section). Now you just present a technical report from architecture and hyperparameters optimization.

5) The manuscript is full of technical information that can be moved to supplementary data (tables 3, 4, 5, 6, 7, 8, 9, 10, 11, 12, 13 - this is a very standard procedure of the architecture and hyperparameters optimization). Similarly, Figures 6-9.

6) Section 390-402 can be removed as it makes no sense as for the classification we do not use MSE, MAE, or RMSE (in consequence I do not understand what is presented in Table 14)

Minor comments:
1) Figure 1 could be moved to the supplement as it describes a very standard workflow in ML design (I do not see here anything new).

2) Do not use jargon e.g. l22-23 "In this paper, ..."

3) Add space after digits e.g. l. 35 "0.00447seconds"

4) Figures 10-13 are missing. In figure 14 remove the 3D effect from the plot (again this should be runtime, not compilation time).

Author Response

The presented manuscript shows some results related to deep learning models for Chronic Kidney Disease prediction. The following concerns need to be addressed:

Major comments:
1) The datasets, ML models, and raw predictions used/presented in this study had not been provided. Without them, it is impossible to validate any of the presented results. Please provide them as supplementary data or/and as a Github repository.

Reply: Here we shared the GitHub repository. Link: https://github.com/Shamrat777/Kidney-Disease
We also added the link to the paper. (line: 534-536)

2) Extensive editing of English language and style required. The text is full of typos (l.249, 250, 253, etc.), and grammar errors (e.g. "Alsuhibany et al., [7] 2021 In this study, they"). The whole text needs to be rewritten, it contains multiple fragments that are badly organized, and some sentences are not needed (trivial information) or are very similar to the previous ones. Some sentences make no sense. For instance, lines 130-136 are hard to read, and the sentence "The dataset is processed to 131 provides a better intuition." make no sense. Then we have repetition related to 80% and 20 % vs 80:20.

Reply: Thank you for your valuable Suggestion. We rewritten and organized related work by your suggestion. We also removed the unnecessary text that doesn’t make any sense (line 70-72).

3) Use proper nomenclature e.g in computer science we do not use compilation time but rather runtime in the context presented in the manuscript

Reply: Thank you for your valuable Suggestion. In compilation time we replace it by runtime (line 472, 473, 479).

4) I do not understand the ML section as it is described now. I do not know in what way the so-called modified versions of CNN, ANN and LSTM were different than the standard ones. The text describes very standard optimization of the number of layers or/and hyperparameters. I do not see here a novel modification of the classical CNN, ANN, or LSTM architectures. The whole description of ML models is very poor. For instance, I do not understand what the RBF kernel has to do to CNN (l.177-180). In figure 3 regarding the CNN network, there is only some MLP model presented (lack of CNN elements). In general, you need to compare your best, optimized versions of CNN, ANN, and LSTM models to the ones designed by others (some of the methods mentioned in the "Related work" section). Now you just present a technical report from architecture and hyperparameters optimization.

Reply: Thank you for your valuable Suggestion. We write RBF by mistake now we correct this part (line 176-179). We have done ablation study on the simple deep learning model by changing different parameters, using some parameters we got the highest accuracy that is the modified version of simple deep learning model which is used in our study (line 172-361). Also, we made a comparison between our work and existing work as per your advice (line 501-509).

5) The manuscript is full of technical information that can be moved to supplementary data (tables 3, 4, 5, 6, 7, 8, 9, 10, 11, 12, 13 - this is a very standard procedure of the architecture and hyperparameters optimization). Similarly, Figures 6-9.
Reply: Thank you for your valuable Suggestion. Without explain of the figures, its not possible to understand the performance of the models.  

6) Section 390-402 can be removed as it makes no sense as for the classification we do not use MSE, MAE, or RMSE (in consequence I do not understand what is presented in Table 14)

Reply: Thank you for your valuable Suggestion. We removed MSE, MAE, or RMSE (389-400). Table 14 was serial mistake now it is in Table 16 and we show evolution of prediction results (false positive rate and false negative rate, negative predictive value and false discovery rates) for the three classifiers (line 482-484).

Minor comments:
1) Figure 1 could be moved to the supplement as it describes a very standard workflow in ML design (I do not see here anything new).
Reply: We made little bit change in our diagram, we removed unnecessary things and tried to represent our workflow hope it is better now.

2) Do not use jargon e.g. l22-23 "In this paper, ..."
Reply: We removed all jargon.

3) Add space after digits e.g. l. 35 "0.00447seconds"
Reply: We correct all of this kind of mistake. Thank you for your suggestion (line 475-477).

4) Figures 10-13 are missing. In figure 14 remove the 3D effect from the plot (again this should be runtime, not compilation time).

Reply: We write figure 10 by mistake as figure 14, we correct figures serial number now. 3D effect is removed.

Reviewer 2 Report

The paper can be improved in at least the methodological parts and second in the simulation study.

a) Methodological Aspects:

a1) Only a single data set is considered, and more important results from the literature are missing in the study. My understanding is that this data has been applied in the literature to some extent and therefore I suggest to introduce these results in the discussion of the results of the performed simulation study.

a2) the proposed methods are all well know, therefore the authors must clearly explain the degree of novelty of their proposed methods.

a3) The description of the CNN (see 179) it is mentioned that an RBF kernel function is used. But this is not displayed in Figure 3. Please check this again. Also in the tables I cannot find something about RBF

a4) besides the Figures 3 , 4, and 5 nothing is discussed on the neural networks architectures. All in all the description of the used architectures is very high level. Again, my question what is the novelty in this regard.

b) Simulation study.

b1) A comparision to other results in the literature is missing

b2) only single results are reported; variance measures are missing.

b3) it seems that only a single trial study has been performed, and so statistical significance testing is missing in the study.

b4) A strong classifier for such a data set of missed feature scales - nominal & metric - would be the Random Forest, do you have results for it.

Minor comments

- in 3.2 the problem of missing data is reported and the fact that data imputation as been applied and the results were more accurate . Please explain this in more detail. How big is the amount of missing data, what is the effect of imputation, why average ...?

- in 3.3. The RBF kernel was mentioned, Here the width parameter is an important parameter. Please explain how this was fixed.

- equation 9 : absolute value missing?

- Figures 6, 7 , 8, 9 : Please skip the meaning less lines in the plots, only the dots are the measurements , right?

Author Response

The paper can be improved in at least the methodological parts and second in the simulation study.

  1. a) Methodological Aspects:

a1) Only a single data set is considered, and more important results from the literature are missing in the study. My understanding is that this data has been applied in the literature to some extent and therefore I suggest to introduce these results in the discussion of the results of the performed simulation study.

Reply: Thank you for your review. After your review we added discussion section of our work, where we mentioned why we avoid important results from the literature in our study (line 490-499). We also avoid feature selection techniques in our study because the feature of our CKD datasets is highly correlated (line 166-171). In future we will work with a merge dataset by applying this hybrid model and cross validation and feature selection techniques (line 527-531).

a2) the proposed methods are all well know, therefore the authors must clearly explain the degree of novelty of their proposed methods.

Reply: Thank you for your valuable Suggestion. In this paper, we proposed three modified deep learning algorithms CNN, ANN, and LSTM, to predict CKD at the primary stage. Multiple layers with linear or nonlinear activation functions are assembled to form the model. The description is already added to the paper (line: 54-58, 112-133)

a3) The description of the CNN (see 179) it is mentioned that an RBF kernel function is used. But this is not displayed in Figure 3. Please check this again. Also in the tables I cannot find something about RBF

Reply: Thank you for your valuable Suggestion. We write RBF by mistake now we correct this part (line 176-179).

a4) besides the Figures 3, 4, and 5 nothing is discussed on the neural networks architectures. All in all the description of the used architectures is very high level. Again, my question what is the novelty in this regard.

Reply: Thank you for your valuable Suggestion. Actually, we employed the standard CNN, ANN, LSTM and proposed MCNN, MANN, MLSTM to get the better accuracy than existing models. Also, the comparison of proposed model with existing work proves that we get highest accuracy. We add layers, optimizer and activation function to CNN, ANN, LSTM to build our proposed models. We already added the description of the models to the paper (line: 194-199, 247-252, 306-310)

  1. b) Simulation study.

b1) A comparison to other results in the literature is missing.

Reply: Thank you for your valuable suggestion. We made a comparison between our work and existing work as per your advice (line 493-501).

b2) only single results are reported; variance measures are missing.

Reply:  Thank you for your review. To check the validity of our model we will performed this model on different datasets in future work (line 530-531).

b3) it seems that only a single trial study has been performed, and so statistical significance testing is missing in the study.

Reply: Thank you for your valuable advice. This work was not in our plan, so it was not done. But we are very happy for your valuable feedback, we will add it in future work to expand the work (line 532-533).

b4) A strong classifier for such a data set of missed feature scales - nominal & metric - would be the Random Forest, do you have results for it.

Reply: Thank you for your valuable suggestion. We did not apply Random Forest Classifier in our datasets. After your suggestion we performed Random Forest Classifier in our dataset. But it did not perform well. It provides lowest accuracy (94.25%) compared to other three proposed model. That’s why we did not add this result in our paper.

Minor comments

- in 3.2 the problem of missing data is reported and the fact that data imputation as been applied and the results were more accurate . Please explain this in more detail. How big is the amount of missing data, what is the effect of imputation, why average ...?

Reply: Thank you for your valuable advice. In our dataset, some feature values were very low and some feature values were very high. Hence the tendency to get inconsistent values was very high. For this reason, we used average to overcome this problem.

- in 3.3. The RBF kernel was mentioned, Here the width parameter is an important parameter. Please explain how this was fixed.

Reply: Thank you for your valuable Suggestion. We write RBF by mistake now we correct this part (line 176-179).

- equation 9 : absolute value missing?

Reply: Thank you for your valuable advice. We already removed this part as per reviewer 1 suggestion (line 388-399).

- Figures 6, 7 , 8, 9 : Please skip the meaning less lines in the plots, only the dots are the measurements , right?

Reply: Thank you for your valuable Suggestion. The lines are to make sure that readers don’t miss the progress of the dots. For that reason, we made the lines thinner.

Round 2

Reviewer 1 Report

Reviewer's Reply R2: The Authors did not address all concerns properly, thus I recommend another round of revision:

Major comments:
1) The datasets, ML models, and raw predictions used/presented in this study had not been provided. Without them, it is impossible to validate any of the presented results. Please provide them as supplementary data or/and as a Github repository.

Reply: Here we shared the GitHub repository. Link: https://github.com/Shamrat777/Kidney-Disease
We also added the link to the paper. (line: 534-536)

Reviewer's Reply R2: Mentioned GitHub repository contains only Jupyter notebook, which is far from providing complete set of files that I requested. The dataset is imported from Google Colab (instead being provided as separate file directly in the project’s github recommended option; alternatively write a wrapper that automatically download it from proper url; do not write that the dataset is available from some reference as you do not know if this file will be available in future if you will not store it by yourself). Currently, all ML models must be re-trained ad hoc. Please write separate training scripts and store trained ML models (after this step ML models must be frozen and never retrained). Than write the prediction scripts loading ML models and do predictions. Store also individual prediction files for each ML model.

----------

2) Extensive editing of English language and style required. The text is full of typos (l.249, 250, 253, etc.), and grammar errors (e.g. "Alsuhibany et al., [7] 2021 In this study, they"). The whole text needs to be rewritten, it contains multiple fragments that are badly organized, and some sentences are not needed (trivial information) or are very similar to the previous ones. Some sentences make no sense. For instance, lines 130-136 are hard to read, and the sentence "The dataset is processed to 131 provides a better intuition." make no sense. Then we have repetition related to 80% and 20 % vs 80:20.

Reply: Thank you for your valuable Suggestion. We rewritten and organized related work by your suggestion. We also removed the unnecessary text that doesn’t make any sense (line 70-72).

Reviewer's Reply R2: The extensive editing of English language, style or even ML terminology is still required (especially in the new fragments; some corrections have been done, but in general the text is still painful to read, for instance there is no such thing as “dropout library”).

-------------

4) I do not understand the ML section as it is described now. I do not know in what way the so-called modified versions of CNN, ANN and LSTM were different than the standard ones. The text describes very standard optimization of the number of layers or/and hyperparameters. I do not see here a novel modification of the classical CNN, ANN, or LSTM architectures. The whole description of ML models is very poor. For instance, I do not understand what the RBF kernel has to do to CNN (l.177-180). In figure 3 regarding the CNN network, there is only some MLP model presented (lack of CNN elements). In general, you need to compare your best, optimized versions of CNN, ANN, and LSTM models to the ones designed by others (some of the methods mentioned in the "Related work" section). Now you just present a technical report from architecture and hyperparameters optimization.

Reply: Thank you for your valuable Suggestion. We write RBF by mistake now we correct this part (line 176-179). We have done ablation study on the simple deep learning model by changing different parameters, using some parameters we got the highest accuracy that is the modified version of simple deep learning model which is used in our study (line 172-361). Also, we made a comparison between our work and existing work as per your advice (line 501-509).

Reviewer's Reply R2: Try also selu, softplus and softsigns (in all combinations) activation layers, especially in CNN architecture (in my experience they worked much better than relu or sigmoid). Additionally, replace ‘modified’ with ‘optimized’ whenever you mention your models (the word ‘modified’ suggest that you introduced completely new type of ML architecture e.g. new version CNN with unusual, mathematically new, building elements which is not the case here). Moreover, please correct Figure 3 as this is not how CNN network looks like.

-------------

5) The manuscript is full of technical information that can be moved to supplementary data (tables 3, 4, 5, 6, 7, 8, 9, 10, 11, 12, 13 - this is a very standard procedure of the architecture and hyperparameters optimization). Similarly, Figures 6-9.
Reply: Thank you for your valuable Suggestion. Without explain of the figures, its not possible to understand the performance of the models.

Reviewer's Reply R2: Again, please move half of the mentioned tables and figures to the supplement and just mention the main results from those experiments in the main text (most of those tables and figures refers to very technical aspects and only very interested readers will expect them while the rest will not require them to understand the main message). Moreover, the order of the tables is wrong (Table 16 appears before Table 15).

-----------------

6) Section 390-402 can be removed as it makes no sense as for the classification we do not use MSE, MAE, or RMSE (in consequence I do not understand what is presented in Table 14)

Reply: Thank you for your valuable Suggestion. We removed MSE, MAE, or RMSE (389-400). Table 14 was serial mistake now it is in Table 16 and we show evolution of prediction results (false positive rate and false negative rate, negative predictive value and false discovery rates) for the three classifiers (line 482-484).

Reviewer's Reply R2: Clean also the code on GitHub from the regression ‘fossils’.

------------------

Author Response

Reviewer-1:

Thanks for accepting our manuscript.

Reviewer-2:

Major comments:

1) The datasets, ML models, and raw predictions used/presented in this study had not been provided. Without them, it is impossible to validate any of the presented results. Please provide them as supplementary data or/and as a Github repository.

Reply: Here we shared the GitHub repository. Link: https://github.com/Shamrat777/Kidney-Disease
We also added the link to the paper. (line: 534-536)

Reviewer's Reply R2: Mentioned GitHub repository contains only Jupyter notebook, which is far from providing complete set of files that I requested. The dataset is imported from Google Colab (instead being provided as separate file directly in the project’s github – recommended option; alternatively write a wrapper that automatically download it from proper url; do not write that the dataset is available from some reference as you do not know if this file will be available in future if you will not store it by yourself). Currently, all ML models must be re-trained ad hoc. Please write separate training scripts and store trained ML models (after this step ML models must be frozen and never retrained). Than write the prediction scripts loading ML models and do predictions. Store also individual prediction files for each ML model.

Reply: Thank you for your valuable suggestion. We have corrected and uploaded the code as per your requirements. Here we shared the GitHub repository.

Link:  https://github.com/robiulxasan/Kidney-Disease.git

We also added the link to the paper. (line:516-517)
----------

2) Extensive editing of English language and style required. The text is full of typos (l.249, 250, 253, etc.), and grammar errors (e.g. "Alsuhibany et al., [7] 2021 In this study, they"). The whole text needs to be rewritten, it contains multiple fragments that are badly organized, and some sentences are not needed (trivial information) or are very similar to the previous ones. Some sentences make no sense. For instance, lines 130-136 are hard to read, and the sentence "The dataset is processed to 131 provides a better intuition." make no sense. Then we have repetition related to 80% and 20 % vs 80:20.

Reply: Thank you for your valuable Suggestion. We rewritten and organized related work by your suggestion. We also removed the unnecessary text that doesn’t make any sense (line 70-72).

Reviewer's Reply R2: The extensive editing of English language, style or even ML terminology is still required (especially in the new fragments; some corrections have been done, but in general the text is still painful to read, for instance there is no such thing as “dropout library”).

Reply: Thank you for your suggestion. We have revised the paper and made necessary corrections.

-------------

4) I do not understand the ML section as it is described now. I do not know in what way the so-called modified versions of CNN, ANN and LSTM were different than the standard ones. The text describes very standard optimization of the number of layers or/and hyperparameters. I do not see here a novel modification of the classical CNN, ANN, or LSTM architectures. The whole description of ML models is very poor. For instance, I do not understand what the RBF kernel has to do to CNN (l.177-180). In figure 3 regarding the CNN network, there is only some MLP model presented (lack of CNN elements). In general, you need to compare your best, optimized versions of CNN, ANN, and LSTM models to the ones designed by others (some of the methods mentioned in the "Related work" section). Now you just present a technical report from architecture and hyperparameters optimization.

Reply: Thank you for your valuable Suggestion. We write RBF by mistake now we correct this part (line 176-179). We have done ablation study on the simple deep learning model by changing different parameters, using some parameters we got the highest accuracy that is the modified version of simple deep learning model which is used in our study (line 172-361). Also, we made a comparison between our work and existing work as per your advice (line 501-509).

Reviewer's Reply R2: Try also selu, softplus and softsigns (in all combinations) activation layers, especially in CNN architecture (in my experience they worked much better than relu or sigmoid). Additionally, replace ‘modified’ with ‘optimized’ whenever you mention your models (the word ‘modified’ suggest that you introduced completely new type of ML architecture e.g. new version CNN with unusual, mathematically new, building elements which is not the case here). Moreover, please correct Figure 3 as this is not how CNN network looks like.

Reply: Thank you for your valuable comment. We tried selu, softplus and softsigns, but the previous results are still better. 
As per your suggestion we replaced the ‘modified’ with ‘optimized’.

There are some mistakes in figure 3, we changed the figure and added to the paper.

-------------

5) The manuscript is full of technical information that can be moved to supplementary data (tables 3, 4, 5, 6, 7, 8, 9, 10, 11, 12, 13 - this is a very standard procedure of the architecture and hyperparameters optimization). Similarly, Figures 6-9.
Reply: Thank you for your valuable Suggestion. Without explain of the figures, its not possible to understand the performance of the models.

Reviewer's Reply R2: Again, please move half of the mentioned tables and figures to the supplement and just mention the main results from those experiments in the main text (most of those tables and figures refers to very technical aspects and only very interested readers will expect them while the rest will not require them to understand the main message). Moreover, the order of the tables is wrong (Table 16 appears before Table 15).

Reply: Thank you for your suggestion. We have moved all of your mentioned tables and figures to the supplementary file as per your suggestion.

We have also checked the table number and corrected.

-----------------

6) Section 390-402 can be removed as it makes no sense as for the classification we do not use MSE, MAE, or RMSE (in consequence I do not understand what is presented in Table 14)

Reply: Thank you for your valuable Suggestion. We removed MSE, MAE, or RMSE (389-400). Table 14 was serial mistake now it is in Table 16 and we show evolution of prediction results (false positive rate and false negative rate, negative predictive value and false discovery rates) for the three classifiers (line 482-484).

Reviewer's Reply R2: Clean also the code on GitHub from the regression ‘fossils’.

Reply: Thank you for your suggestion. We cleaned the code from the regression ‘fossils’ and uploaded it to GitHub. We also added the GitHub repository link to the paper. (Line:515-516).

Reviewer 2 Report

Thanks for the revision - I have got no further comments

Author Response

Thanks for accepting our manuscript.